# Polycotton waste textile recycling by sequential hydrolysis and glycolysis

Nienke Leenders[1], Rijk M. Moerbeek[2], Matthijs J. Puijk[3], Robbert J. A. Bronkhorst[3], Jorge Bueno Morón[3], Gerard P. M. van Klink [1,3] & Gert-Jan M. Gruter [1,3] ✉

As a result of the current high throughput of the fast fashion collections and the concomitant decrease in product lifetime, we are facing enormous amounts of textile waste. Since textiles are often a blend of multiple fibers (predominantly cotton and polyester) and contain various different components, proper waste management and recycling are challenging. Here, we describe a high-yield process for the sequential chemical recycling of cotton and polyester from mixed waste textiles. The utilization of 43 wt% hydrochloric acid for the acid hydrolysis of polycotton (44/56 cotton/polyester, room temperature, 24 h) results in a 75% molar glucose yield from the cotton fraction, whereafter the hydrolysate solution is easily separated from the solid polyester residue. The reaction is scalable, as similar results are obtained for experiments performed at 1 mL, 0.1, and 1.0 L and even in a 230 L pilot plant reactor, where mixed postconsumer polycotton waste textile is successfully recycled. The residual polyester is successfully converted via glycolysis to bis(2-hydroxyethyl) terephthalate in 78% isolated yield (>98% purity).

As a result of the current high throughput of the fast fashion collections and the concomitant decrease in product lifetime, we are facing enormous amounts of textile waste[1]. Since textiles are often a blend of multiple fibers (predominantly cotton and polyester) and contain various different components, proper waste management and recycling are challenging[1-4]. Globally, textile fiber production reached a record of 113 million tons in 2021, and it is expected to further expand to 149 million tons by 2030[5]. With a recycling rate of less than 1%, the textile industry is the third most polluting industry, directly after the oil and gas industry and agriculture industry[6,7]. The low recycling rate is a direct result of the complexity of the material. The downcycling of textiles into low-value materials such as sofa stuffing, car insulation or carpet padding is currently the best alternative to incineration and landfilling[8]. Proper recycling is necessary for the textile sector to transition into a true circular and low-carbon industry. Additionally, the introduction of the extended producer responsibility (EPR), which makes the textile producers financially and/or physically responsible for the disposal or treatment of postconsumer waste textiles, increased interest in recycling textile waste to an all-time high[7,9].

Considerable research has been devoted to recycling of polycotton waste, the most common fraction of textile waste, as well as possible reuse applications[1,4,10-18]. To date, no process with a viable business case has been developed in which both polyester and cotton fractions can be effectively fractionated and recycled in high yields. For effective recycling of polycotton waste textiles, a complete separation and utilization of the different components is required. Depolymerization of cotton via acid hydrolysis has been reported in the literature; nonetheless, most related research for polycotton waste has focused on the use of a low acid concentration at high temperature or on using a multistep process with different acid concentrations[3,19-22]. In contradiction to lignocellulosic biomass, little is known about superconcentrated HCl acid hydrolysis of textiles at ambient temperature. Technology company Avantium developed the DAWN Technology, a modern version of the Bergius process originated and commercialized in the 1920–40s; this technology can depolymerize polysaccharides from lignocellulose, such as wood, into their constituent sugars[23,24]. Several chemical recycling processes focusing on cotton recycling from (poly)cotton textile were evaluated at pilot scale (Supplementary

[1]Van 't Hoff Institute for Molecular Sciences, University of Amsterdam, Amsterdam, The Netherlands. [2]Faculty of Science and Technology, Hogeschool Leiden, Leiden, The Netherlands. [3]Avantium Support BV, Amsterdam, The Netherlands. ✉e-mail: g.j.m.gruter@uva.nl

Information: Table S1), however, none can obtain a high glucose yield from the cotton fraction and leave the poly(ethylene terephthalate) (PET) unaffected. By using superconcentrated HCl on polycotton waste, a glucose solution is obtained from cellulose hydrolysis, which can be easily separated from the solid polyester residue providing the ability to fully recycle both components in subsequent steps. The acidic glucose solution can be used for the production of 5-(chloromethyl)furfural (CMF), a process Avantium is developing for the production of 2,5-furandicarboxylic acid (FDCA)[25,26]. By subsequently converting the glucose into CMF in a biphasic reaction[25], the challenging and costly acid sugar separation can be avoided which immensely decreases the production cost of such a recycling process.

## Results and discussion

### Material analysis

As roughly two-fifths of the labels in textiles contain incorrect information[27], the postconsumer waste textile used in this study was analyzed for its cotton/polyester ratio via thermogravimetric analysis (TGA), the National Renewable Energy Laboratory (NREL) protocol[28], and by Celignis Analytical using an acid hydrolysis protocol[29]. TGA (Fig. 1a) showed that the textile consisted of one minor and two major components. According to the derivative thermogravimetric (DTG) data (Fig. 1b), the first major decomposition peak at a peak temperature of 321 °C corresponds to cotton, and the second decomposition peak at a peak temperature of 411 °C corresponds to polyester. Both decomposition temperatures were confirmed with pure reference materials. The small decrease (3%) in mass observed before 100 °C corresponds to the water held by the cotton evaporating from the textile. Additionally, the DTG curves were deconvoluted to obtain a cotton/polyester ratio of 44/56, resulting in a composition of 43% cotton, 54% polyester (sum of squared errors = 6.56 × 10⁻⁶), and 3% water in the waste textile. The NREL protocol was performed in sextuplo, and it was found that the waste textile contained 44% cotton

(SD 3%). After cotton hydrolysis, the residual materials were dried, weighed and analyzed to find the material contained 54% polyester (SD 2%). These results are also in line with the results from Celignis Analytical (cotton content 44% (SD 0.6%)).

Scanning electron microscopy (SEM) images were acquired to obtain additional information on the structure of the yarn. In Fig. 1c, a clear difference can be seen between the polyester and cotton fibers. The polyester fiber has a cylindrical shape and a smooth surface, whereas the cotton fiber has a twisted kidney shape and a rough surface[30]. Zoomed out (Fig. 1c) shows that the yarns are twisted bundles of polyester and cotton fibers that are weaved together.

### Acid hydrolysis with different HCl concentrations

At ambient temperature (15–24 °C), superconcentrated hydrochloric acid is necessary to disrupt the recalcitrant crystalline structures formed by the inter- and intramolecular hydrogen bonds between the cellulose hydroxyl groups in cotton[31]. Initially, hydrolysis was performed with 37 or 43 wt% aq. HCl to investigate the effect of acid concentration on glucose yield and on the formation of the glucose dehydration/degradation products 5-(hydroxymethyl)furfural (HMF), levulinic acid and humins at given reaction times (Supplementary Information: Fig. S1)[32–34]. The formation of HMF from glucose can follow a cyclic or acyclic reaction mechanism, both with and without fructose isomerization[35,36]. HMF can further degrade to levulinic acid and formic acid (1:1) and to humins (undesirable, carbon-based macromolecules), which are formed via the polymerization of glucose with HMF[37,38]. When assessing the glucose and byproduct yields after hydrolysis at different reaction times (Fig. 2a), 37 wt% aq. HCl after 3 days only led to a moderate glucose yield (25%) and low HMF formation (<0.5%) (Fig. 2b). The levulinic acid concentrations were low and could not be properly quantified due to a signal-to-noise ratio that is too low. The glucose yield seemed to increase slightly with increasing reaction time up to three days (25%), after which the glucose

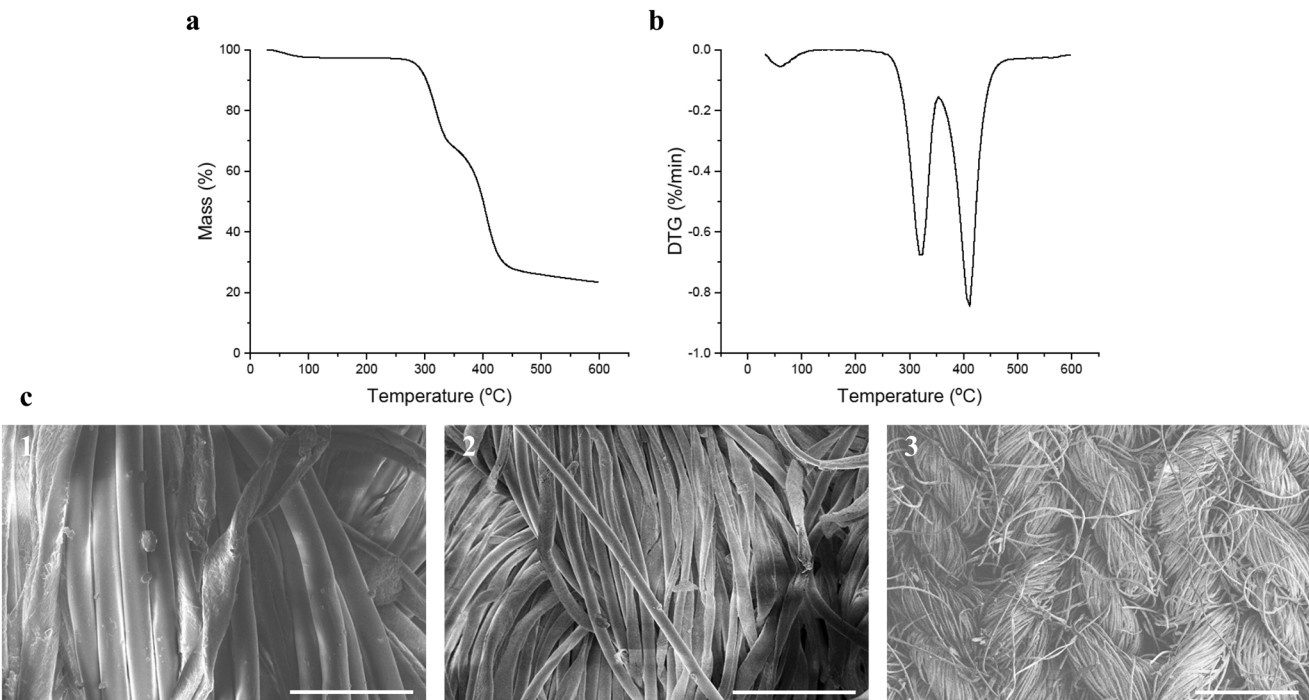

**Fig. 1 | Material analysis of the polycotton waste textile samples.**
**a** Thermogravimetric analysis (TGA) results of the polycotton waste textile before acid hydrolysis (heating rate 2.5 °C/min under N₂ flow). **b** Derivative thermogravimetric (DTG) results of the polycotton waste textile before acid hydrolysis (heating rate 2.5 °C/min under N₂ flow). The first major peak corresponds to cotton, and the second major peak corresponds to polyester. **c** Scanning electron microscope (SEM) images of the polycotton waste textile before acid hydrolysis at 1200×, 600×, and 100× magnification, respectively. At 600× and 1200× magnification, the polyester and cotton filaments can be clearly distinguished. Scale bars = 50 μm, 100 μm, 500 μm, respectively.

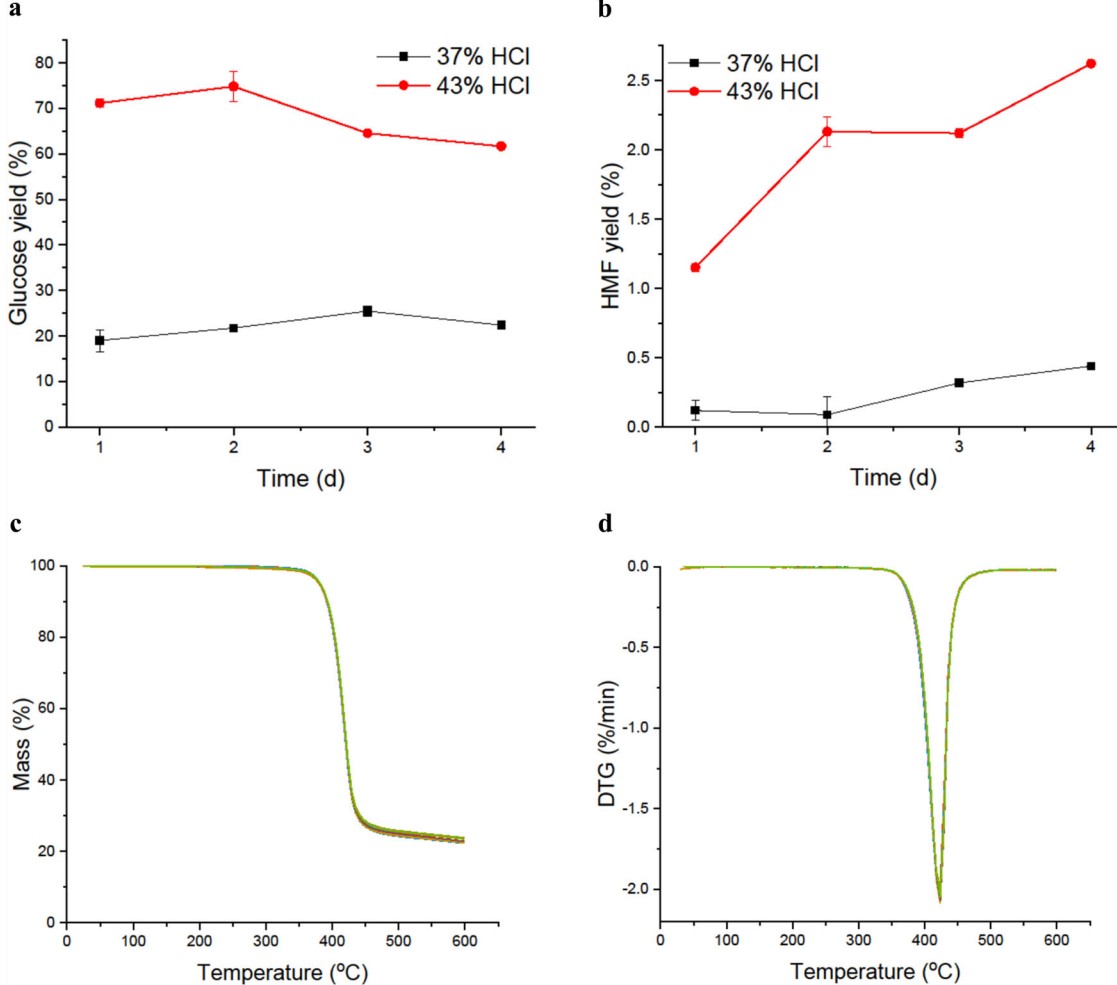

**Fig. 2 | Acid hydrolysis of 44/56 polycotton waste textiles. a** Glucose yield after hydrochloric acid (HCl) hydrolysis (50 mg/mL, 1000 rpm) of waste textiles over time in days (d). Shown values are averaged from three measurements and the error bars indicate the standard deviation. **b** 5-(Hydroxymethyl)furfural (HMF) yield after HCl hydrolysis (50 mg/mL, 1000 rpm) of waste textile over time in days (d). Shown values are averaged from three measurements and the error bars indicate the standard deviation. **c** TGA result of the residual waste textile after hydrolysis with 43 wt% aq. HCl for 1–4 days (heating rate 2.5 °C/min under $N_2$ flow). All 12 measured samples showed only one decrease in mass and were identical to one another. **d** DTG result of the residual waste textile after hydrolysis with 43 wt% aq. HCl for 1–4 days (heating rate 2.5 °C/min under $N_2$ flow). The peak corresponds to polyester degradation.

yield started to decrease. With 43 wt% aq. HCl, considerably higher glucose yields (70–75%) were obtained in a one-step synthesis, which, to our knowledge, has not yet been reported for cotton from concentrated acid hydrolysis. Costa et al. reported cellulose extraction from preconsumer polycotton textiles via one-step acid hydrolysis with 1 mol/L nitric acid under reflux for 7 h, which led to an extraction yield of 26% cellulose[20]. Sanchis-Sebastiá et al. reported an unsuccessful attempt to obtain high glucose production by a one-step synthesis[19]. They were able to obtain a 90% glucose yield via two-step acid hydrolysis with 72% $H_2SO_4$ in 100% cotton bed sheets, which is the highest glucose yield reported. However, the use of sulfuric acid requires an additional acid sugar separation step, which has been proven to be a bottleneck. Using 43 wt% aq. HCl, a somewhat faster decline in glucose yield over time was noted compared to that of the hydrolysis using 37 wt% aq. HCl. Additionally, a greater increase, albeit still low, in HMF yield over time was observed (1–2.5%), which in this case would not be a problem as HMF can be converted to CMF in the subsequent reaction step.

As there is a significant difference between the molar glucose yields obtained by hydrolysis with 37 and 43 wt% aq. HCl, the textile solid residues after hydrolysis were analyzed by TGA. For textiles subjected to hydrolysis with 37 wt% aq. HCl (Supplementary Information: Fig. S2), not all the cotton was removed, indicating incomplete hydrolysis. However, no direct correlation was detected between the remaining cotton content in the waste textile and the reaction time, which could indicate the presence of more or less recalcitrant cellulose in cotton. This phenomenon is not uncommon as Higgins and Ho found that 37 wt% HCl is also insufficient when hydrolyzing cellulose from wood and wood waste[39].

The TGA results of the hydrolysis with 43 wt% aq. HCl (Fig. 2c) revealed complete hydrolysis for all the samples, irrespective of the reaction time, as only the decomposition peak of polyester was seen for the residues. The mass of the residual textile after cotton hydrolysis was found to be comparable to the measured polyester fraction. The peak decomposition temperature of the polyester residue of 423 °C is slightly higher compared to the peak decomposition temperature of polyester in the blend (Fig. 2d). This change in thermal properties in blended fabrics has been mentioned before by Chen and Zhao[18]. They found that cotton char promotes the combustion of polyester and decreases the peak heat release rate temperature compared to a pure polyester sample. As the peak heat release temperature and peak decomposition temperature closely match when the mass pyrolyzed is combustible[40], it is probable that the slight decrease in peak decomposition temperature was promoted by the cotton char.

In all the small-scale hydrolysis reactions (1 mL), the formation of humins was observed. However, due to the limited scale of the experiments, precise quantification of the humins was not feasible. It is presumed that the reactors containing 43 wt% aq. HCl had a higher humin content than did those with 37 wt% aq. HCl, as evidenced by the darker color and increased opacity of the solutions (Supplementary Information: Fig. S3).

Additionally, an investigation into the impact of a posthydrolysis step (1 wt% HCl, 120 °C, 1 h, 1000 rpm) was performed. It was found that samples initially subjected to hydrolysis with 37 or 43 wt% aq. HCl treatment increased the molar glucose yield by 23 and 11% after posthydrolysis, respectively, suggesting the presence of glucose oligomers in the hydrolysates.

Considering all the results, 37 wt% aq. HCl is not sufficient for reliable quantitative cotton removal at ambient temperature. Although the formation of byproducts was slightly lower than that of superconcentrated HCl, the inconsistent removal of cotton inhibited the use of 37 wt% aq. HCl as full separation is required for PET recycling, as any cotton that remains severely hinders the polyester recycling process. These results are similar to those observed and reported for the hydrolysis of old Bergius wood process[23].

### Optimization of the hydrolysis conditions

Although the hydrolysis of polycotton waste textile with 43 wt% aq. HCl resulted in good glucose yields, the reaction was further investigated to determine which improvements could be made. When considering mechanical agitation, no clear trend was observed between the different stirring speeds and no stirring (Fig. 3a). TGA confirmed that all the measured stirring speeds led to complete cotton removal as the residual textile samples contained only polyester (Supplementary Information: Fig. S4). As shown in Fig. 3a, all the conditions led to a decrease in glucose yield over time. Therefore, the hydrolysis with 43 wt% aq. HCl and 50 mg/mL textile loading was monitored during the first 24 h to obtain further insights into the reaction (Fig. 3b). The curve indicated that the glucose yield seemed to reach its optimum at 18 h, after which the curve flattened. The DTG curve of the solid residue after 3 h showed full cotton removal (Supplementary Information: Fig. S5). However, as the glucose yield at 3 h was 22%, large quantities of (dissolved) oligomers were expected to be present. Posthydrolysis led to a high glucose yield of 80%, indicating that a reaction time of 3 h is sufficient to fully separate cotton and polyester but minimizes the decomposition of glucose to HMF and humins compared to a 24 h reaction time.

Considering that the impact of mechanical agitation appears to be negligible, stirring was excluded from the following experiments to design a setup that can be used in the Avantium Dawn Technology pilot plant, which is equipped with packed bed (non-stirred) reactors.

TGA confirmed the complete hydrolysis of cotton from the textiles with a loading of 50 mg/mL and 43 wt% aq. HCl, the loading was subsequently increased to explore the boundary conditions for full cotton hydrolysis. The HPLC results (Fig. 3c) revealed that increasing the textile loading from 50 to 300 mg/mL led to an overall decrease in glucose and HMF formation from 75 to 45% and from 1 to 0.25%, respectively. This aligns with expectations, as higher textile loading leads to a lower acid ratio per mole of cellulose, decreasing the rate of reaction and reducing the formation of byproducts. Additionally, the formation of oligomers in the presence of an acidic catalyst is favored at high glucose concentrations due to the equilibrium between monomeric and oligomeric glucose. This leads to a seemingly overall lower glucose yield. TGA of the residual materials after hydrolysis indicated full cotton removal for all the textile samples (Supplementary Information: Fig. S6). To confirm the presence of large quantities of oligomers, posthydrolysis experiments were performed for the various textile loadings. The results showed consistent glucose yields of 75–80% after posthydrolysis, confirming the presence of large quantities of glucose oligomers.

### Large-scale hydrolysis

Subsequently, the scalability of the reaction was investigated. Hydrolysis was performed at 1 and 100 mL, as well as at 1.0 and 230 L scales, with a textile loading of 50 mg/mL 43 wt% aq. HCl for 24 h in a static closed environment. For all the laboratory experiments (1, 100, and 1000 mL), TGA of the solid residue confirmed complete cotton removal (Supplementary Information: Fig. S7), with similar glucose and HMF yields of 75 and 1%, respectively (Supplementary Information: Fig. S8).

Further scale-up was performed in Avantium's DAWN Technology pilot plant. A 230 L polyvinyl chloride (PVC) reactor was filled with 11.6 kg postconsumer 44/56 polycotton waste textile (for a solid loading of 50 g/L HCl) after which the reactor was filled from the bottom with 43 wt% aq. HCl (-12 °C) at an addition rate of -28 L/h. After 24 h of standing, the reactor was drained thus separating the hydrolysate (-17 °C) from the solid residual material. This material was washed several times with water inside the reactor and afterwards washed in a standard home appliance laundry module. The material in- and outputs, reaction parameters and glucose yields can be found in the Supplementary Information (Table S2). The first trial on pilot plant

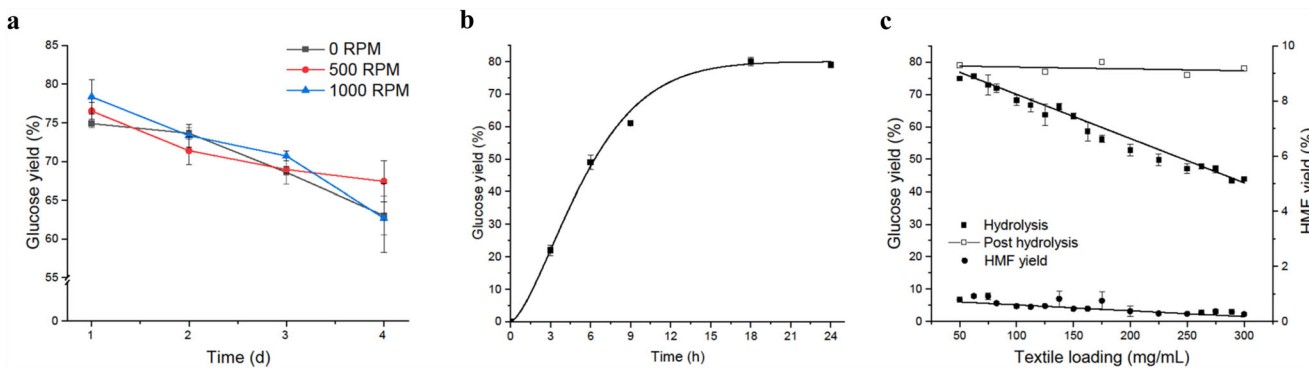

**Fig. 3 | Optimization of the acid hydrolysis conditions. a** Effect of the stirring rate on the glucose yield after hydrolysis (50 mg/mL, 43 wt% aq. HCl) of 44/56 waste polycotton over time (d). Shown values are averaged from three measurements and the error bars indicate the standard deviation. **b** Screening of 44/56 waste polycotton acid hydrolysis over time (50 mg/mL, 43 wt% aq. HCl, static reaction, 0–24 h). Shown values are averaged from three measurements and the error bars indicate the standard deviation. **c** Glucose and HMF yield after hydrolysis (43 wt% aq. HCl, 1 day, static reaction) of 44/56 waste polycotton at different textile loadings. Posthydrolysis leads to very similar overall glucose yields for all loadings (50–300 mg 44/56 polycotton/mL acid). Shown values are averaged from three measurements and the error bars indicate the standard deviation.

scale resulted in a glucose yield of 56% (run 1). The yield is lower compared to the values obtained at laboratory scale (70–75%), however, full cotton removal was achieved making the remaining polyester suitable for subsequent glycolysis (Supplementary Information: Fig. S9). When comparing the reaction parameters of the lab experiments with the pilot plant run, it was found that the temperature of the HCl solution in the pilot plant was slightly lower compared to the temperature in the lab.

Meanwhile, as manual size reduction of the 10s of kgs of pilot plant waste textile materials was very labor intensive, the effect of the available surface area was investigated. A comparison was made between an intact piece of textile and a fully cut piece of similar mass (Supplementary Information: Fig. S10). The effect of the increase in surface area on the glucose yield was minimal (1.3 and 3% increase) at textile loadings of 50 mg/mL and 300 mg/mL, respectively (Supplementary Information: Table S3). Because the effect on yield was minimal, shredding of waste textiles at the pilot plant scale was regarded as unnecessary.

Furthermore, to better understand the glucose yield and possible deviations at the pilot plant scale, a run was performed using pure cotton shirts. A PVC reactor was filled with 11.8 kg of pure cotton shirt (60 shirts) (for a solid loading of 50 g/L HCl), and the same procedure was used as in the first run. After hydrolysis of the pure cotton shirt, only the polyester seams and labels (0.21 kg) were left in the reactor (Supplementary Information: Fig. S11 and Fig. S12). The hydrolysis of pure cotton shirts at the pilot plant scale resulted in a glucose yield of 75% (run 2), which in this case included both the glucose and the disaccharide (cellobiose) quantities produced during the reaction.

As the results are similar to our laboratory findings, a third run was performed with presorted mixed postconsumer polycotton waste textiles (~65% cotton on average) supplied by Wieland Textiles, which were automatically sorted using near-infrared spectroscopy[41]. The reactor was filled with 26.1 kg of waste textile, and the reaction time was increased to 48 h to compensate for the lower reaction rate due to lower HCl solution temperature. After hydrolysis, the residual mixed waste textiles were removed as intact pieces of clothing, which could still be identified as clothing (Supplementary Information: Fig. S13). This reaction also resulted in a glucose yield of 75%, indicating that at the pilot scale, it is feasible to obtain a high yield. This trial did not lead to full separation of cotton from polyester for every garment (Supplementary Information: Fig. S14), as a few pieces of clothing were not completely hydrolyzed. This could be the result of the large increase in

the textile mass, which, in a tightly packed, non-stirred reactor, could lead to air pockets in which not all the cotton is accessible for HCl. This problem is expected to be solved when performing the reaction in a continuous manner (simulated moving bed operation). A number of PVC reactors can be connected in such a way that a counter-current flow of HCl can hydrolyze the stationary textile waste. This would also allow for an increase in glucose concentration, making such a process more economically viable.

As the original DAWN Technology process used lignocellulosic biomass as feedstock, a two-stage process was required to separate hemicellulose-based sugars from cellulose-based glucose[24]. Since cotton contains only cellulose (i.e., glucose), a one-stage process is sufficient to hydrolyze the polycotton waste materials, notably simplifying the process. Additionally, the process no longer includes the valorization of lignin but residual PET, which is seen as a valuable waste stream. The hydrolysate obtained for this process is an excellent starting point for the production of CMF in a biphasic system, which avoids the costly acid sugar separation required for obtaining isolated sugars. CMF is converted to 5-(methoxymethyl)furfural (MMF) which is oxidized to FDCA, the monomer for poly(ethylene furanoate) (PEF), the biobased alternative to PET[25,26,42]. A techno-economic analysis (TEA) was performed, showing that a process starting from polycotton waste is potentially the lowest cost route for producing FDCA[43]. The TEA, summarized at the final section of the Supplementary Information, provided a cost price for MMF between €1000/ton (at 100 kt/year polycotton input scale) to €5000/ton (at 10 kt/year polycotton input scale), which is significantly lower than the transfer price of fructose or hardwood based MMF at the same scale (~2 ton fructose required per ton MMF). Thus, recycling cotton from polycotton waste textile via acid hydrolysis and subsequently converting the glucose to MMF via CMF, thereby liberating polyester for subsequent closed-loop recycling, is a viable approach for recycling these waste textile materials, making this process groundbreaking.

### Polyester recycling

Hydrolysis at a larger scale provided sufficient PET residue for additional analyses. The material was subjected to SEM analysis, and a clear difference was found in the material before and after hydrolysis. As shown in Fig. 4a, cotton is no longer present, which leads to a loosening of the yarns, as they now consist solely of polyester fibers. This could also be observed when holding the material before and after reaction against a light source, as after hydrolysis, the material has greater light transmission (Supplementary Information: Fig. S15).

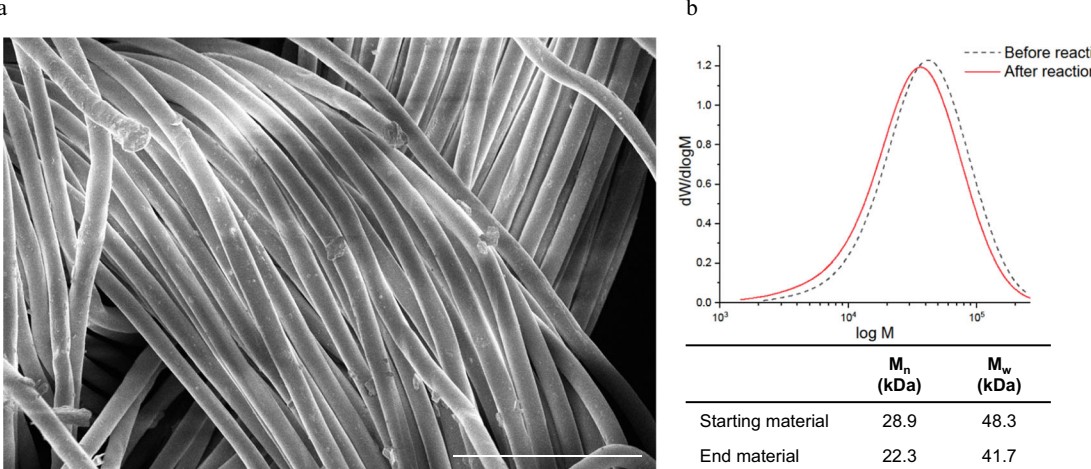

**Fig. 4 | Polyester recycling. a** SEM images of waste textiles after acid hydrolysis. Scale bar = 100 μm. **b** Differential molar mass distributions of poly(ethylene terephthalate) (PET) fibers before and after acid hydrolysis obtained with gel

permeation chromatography (GPC). Number average molar mass ($M_n$) and mass average molar mass ($M_w$) of PET before and after acid hydrolysis.

In addition to TGA and SEM, gel permeation chromatography (GPC) analysis was performed to investigate whether the polyester fibers degraded due to the severe acid hydrolysis conditions during the cotton hydrolysis process. The GPC results (Fig. 4b) showed that PET was slightly hydrolyzed as the $M_n$ and $M_w$ decreased from 28.9 kDa and 48.3 kDa to 22.3 kDa and 41.7 kDa, respectively. This indicates that some polyester hydrolysis occurred during the process. A similar trend in molar mass distribution was found by Haslinger et al., who investigated the production of new man-made cellulose fibers from polycotton waste textiles[15].

However, as changes in molecular weight are believed to have a small effect on the physical properties of residual PET, it is expected that these materials would be fit for mechanical recycling, as a solid-state polymerization step during PET mechanical recycling can bring the molecular weight back to target values. However, because pigments are still present in the polyester residue, it was investigated whether the polyester residue would be suitable for chemical recycling. It must be emphasized that polyester recycling itself is not novel[2,16], however, the experiment was merely performed to confirm that the material is suitable for chemical recycling so the full textile waste stream can be valorized. Therefore, residual PET was depolymerized using glycolysis by refluxing in an excess of ethylene glycol (EG) (molar ratio PET: EG 1:16) and 1 wt% zinc acetate catalyst for 11 h, leading to a bis(2-hydroxyethyl) terephthalate (BHET) isolated yield of 78%. $^1$H NMR spectroscopy (Supplementary Information: Fig. S16) confirmed the production of BHET and gave a purity of 98%[17]. Similar results with our polyester residue were obtained by CuRe Technology, a Dutch polyester rejuvenation company that is scaling up their proprietary PET glycolysis recycling technology. CuRe was able to use these PET residues to produce BHET of sufficient quality for repolymerization to recycled PET, creating a closed-loop recycling process for PET.

## Methods

### Materials
Blue postconsumer polycotton 44/56 cotton/polyester waste textiles were supplied by the Dutch workwear company Groenendijk Bedrijfskleding. Presorted mixed postconsumer polycotton waste textiles (65/35 cotton/polyester) were supplied by Wieland Textiles. The materials were sorted with Fibersort automated sorting technology, which uses near-infrared spectroscopy to sort waste textiles. Pure white cotton shirts were bought at a Dutch chain store. A hydrochloric acid solution in water (37 wt%) was purchased from Fisher Scientific Solvents. Sulfuric acid (95.0–98.0%) was purchased from Thermo Scientific. 1,1,1,3,3,3-hexafluoro-2-propanol (HFIP) was purchased from Biosolve. EG, zinc acetate, deuterated dimethyl sulfoxide (DMSO-$d_6$), and 4-methoxyphenol were purchased from Sigma Aldrich. A super-concentrated HCl solution in water (43 wt%) was produced by absorbing 100% HCl gas in a 35–36 wt% HCl solution. The absorption was performed at 6 °C at atmospheric pressure in an adsorption column (a carbon steel column with PTFE lining) with an absorbing chamber. The HCl gas entered the column from the bottom, the HCl solution reached the top of the column, and the HCl solution was recirculated until the desired HCl concentration was reached. The highly concentrated HCl was cooled and stored at 6 °C at atmospheric pressure. The HCl concentration was determined using a 716 DMS Titrando unit (Metrohm AG) with 1.0 M NaOH at 25 °C. We want to emphasize the difference in concentration between the terms "concentrated HCl" (37 wt%) and "superconcentrated HCl" (43 wt%) used in this article to prevent confusion and misinterpretation of the results.

### (Post)hydrolysis of waste textiles
To convert cotton from the waste materials into glucose, acid hydrolysis was performed. Fifty milligrams of textile and 1 mL of aqueous HCl (37 or 43 wt%) were added to a 9 mL Ace pressure-rated tube (solid loading of 50 mg/mL HCl solution). Reactions were performed at room temperature (21 °C) for 1–4 days with vigorous magnetic stirring (1000 rpm) at autogenous pressure. The hydrolysate was analyzed with IC and HPLC using a calibrated method for carbohydrate quantification, and posthydrolysis was performed by diluting the solution to 1% HCl and heating it at 120 °C for 1 h while mixing at 1000 rpm. All the experiments were performed in triplicate.

### Optimization of hydrolysis conditions
The hydrolysis reaction conditions were further optimized by investigating the effect of mechanical agitation and solid loading. It was found that 43 wt% aq. HCl treatment resulted in a significantly greater glucose yield, therefore, only 43 wt% aq. HCl was used from hereon. Thus, the experiments were performed under similar reaction conditions while the stirring speed was changed to 0 or 500 rpm. These experiments showed that similar glucose yields were obtained even without stirring. To investigate the effect of textile loading, an experiment was also performed by increasing the textile loading up to 300 mg/mL HCl solution at room temperature (21 °C) without stirring for 1 day. All the experiments were performed in triplicate.

### Large-scale hydrolysis
As relatively high glucose yields were obtained during small-scale hydrolysis, the experiment was scaled to a 100 mL HCl solution. The experiment was performed using similar reaction conditions and thus hydrolysis at room temperature (21 °C) with a solid loading of 50 g/L HCl solution for 1 day without stirring. Since the results were comparable to those of the small-scale experiment, the reaction was further scaled to a 1 L HCl solution with a solid loading of 50 g of textile/L HCl solution.

### Pilot plant trial
A further scale-up was performed in Avantium's DAWN Technology pilot plant. A 230 L polyvinyl chloride reactor was filled with textile, after which the reactor was filled with 43 wt% aq. HCl. Afterwards, the reactor was drained, thus separating the hydrolysate from the residual material. The material was washed several times inside the reactor and subsequently washed in a standard home appliance laundry module.

### Recycling of polyesters obtained after hydrolysis of waste textiles
By performing hydrolysis at a larger scale, more polyester residue was generated, which provided the possibility of not only analyzing the residue but also chemically recycling the polyester. The residual polyester was depolymerized using glycolysis with an excess of EG (1:16 weight ratio PET: EG) and 1 wt% zinc acetate while refluxing for 11 h. Afterwards, the reaction mixture was filtered and the filtrate was dissolved in boiling water and filtered again, after which the filtrate was cooled to obtain the crude solid BHET. The purity of the obtained BHET was determined by proton nuclear magnetic resonance ($^1$H NMR) spectroscopy using a 4-methoxyphenol standard.

### Analytical methods
**NREL analysis.** The cellulose and thus cotton content in the polycotton waste textile was analyzed using the NREL protocol for the determination of structural carbohydrates[28]. However, the desired mass was not milled but was directly placed in the reactor.

**High-performance liquid chromatography (HPLC).** Glucose, HMF, and levulinic acid produced during acid hydrolysis were quantified by using high-performance liquid chromatography (HPLC) with a UV/Vis detector. The components were separated with an Aminex HPX-87H Ion Exclusion column operating at 50 °C using 0.6 mL/min of 5 mM $H_2SO_4$ as the eluent. The yield was calculated based on the initial quantity of glucose moles present in the cotton fraction of the sample.

**Thermogravimetric analysis (TGA).** To determine whether complete cotton removal occurred, the textile materials before and after hydrolysis were subjected to TGA. The analysis was performed on a Mettler Toledo TGA/DSC 3+ from 25 to 600 °C with a heating rate of 2.5 °C/min under nitrogen and a flow rate of 80 mL/min. Before TGA, the samples were washed to remove residual HCl and dried overnight at 110 °C. Approximately 10 mg of sample was used per measurement.

**Gel permeation chromatography (GPC).** To assess the quality of the polyester fibers before and after hydrolysis, GPC was performed to determine the $M_n$ and $M_w$. GPC was performed on an Agilent 1260 Infinity. The samples were dissolved in HFIP at an injection volume = 50 μL, flow rate = 1 mL/min, and run time = 55 min. The system was calibrated with poly(methyl methacrylate).

**Scanning electron microscopy (SEM).** The morphology of the textile before and after hydrolysis was analyzed by SEM. The samples were analyzed with a Fei Verios 460 scanning electron microscope and were sputter-coated with a 50 nm titanium layer before analysis.

**Proton nuclear magnetic resonance ($^1$H NMR).** The BHET was analyzed with proton nuclear magnetic resonance ($^1$H NMR). $^1$H NMR spectroscopy was performed with a Bruker Avance AV400 spectrometer with a Z-gradient high-resolution probe. The sample was dissolved in DMSO-d$_6$.

## Data availability

The authors declare that the data supporting the findings of this study are available within the article and its Supplementary Information file. All other relevant source data is available from the corresponding author upon request.

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

## Acknowledgements

This publication is part of the project MIWATEX (project number [KICH1.ED01.20.006]) of the research program Circularity (KIC), which is (partly) financed by the Dutch Research Council (NWO). Additionally, the authors thank Dr. Bing Wang for help with the glycolysis of PET, Igor Hoogsteder from AMOLF for the collaboration to use the scanning electron microscope, Ramon Pragt from CuRe Technology for the larger scale glycolysis of PET, Dr. Michaël Breedveld for the HPLC analysis of the pilot plant runs, Groenendijk Bedrijfskleding and Wieland Textiles for the supply of the polycotton waste textiles, Hank Vleeming and Wei Zhao of Process Design Center, for performing the Conceptual Process Design and TEA, and Modint for sharing their knowledge on textile recycling.

## Author contributions

N.L. devised the experimental program and wrote the manuscript. N.L. and R.M. performed the experimental work at the laboratory scale. M.P. and R.B. performed the experimental work at the pilot plant. J.B.M., G.v.K., and G.-J.G. conceived the concept.

## Competing interests

In March 2022, J.B.M., G.v.K., and G.-J.G. filed the WO2023/166122 A1 patent for "Method of extracting 5-CMF with an organic solvent from cellulosic fibers and man-made non-cellulosic fibers hydrolyzed together with hydrochloric acid". G.v.K., J.B.M., M.P., and R.B. work at Avantium. G.-J.G. is the CTO of Avantium and holds the chair of Industrial Sustainable Chemistry at the University of Amsterdam. The remaining authors declare no competing interests.
