## [Peer Review file · Nature Communications]

Polycotton waste textile recycling by sequential hydrolysis and glycolysis

Corresponding Author: Professor Gert-Jan Gruter

Version 0:

Reviewer comments:

Reviewer #2

(Remarks to the Author)

I have gone through the manuscript "Polycotton waste textile recycling by sequential hydrolysis and glycolysis" (NCOMMS-24-73235-T) by Leenders et al. and also their rebuttal that they have submitted in response to the comments of Reviewer #3. After careful consideration of the issues raised by Reviewer #3 I stand by my original recommendation to accept this manuscript without further revision. I looked again at the original submission (2024-05-09697) and believe that the authors did address the reviewers' comments thoroughly. As I mentioned already in my previous statement, the authors had provided results of a Conceptual Process Design (CPD) and Techno-Economic Assessment (TEA). This clearly adds a new facet to this study, increasing its relevance as a whole. I must admit that I am not an expert in TEA and cannot assess if all numbers are reasonable. But experts in the field can certainly use this TEA and recalculate easily if some of the reported input data is identified as inaccurate.

The issues addressed by Reviewer #3 could be rightfully considered as shortcomings of the study, but further investigations would exceed the scope of this work, and add only minor additional insight to this already comprehensive investigation. That being said, I want to briefly comment on the issues that were raised by Reviewer #3.

1) I interpret the Table that the authors provided as an effort to show the breath of ongoing chemical textile fiber recycling initiatives. Certainly, it would add some value if the authors had compared their method with other reported procedures that depolymerize cellulose into glucose. Then again, most of the published reports are on laboratory-scale, so they lack data that could be compared with the large-scale trials reported in Leenders et al.

2) The twist is indeed important. But one could also argue that dye stuff, spin finish, heavy metals, etc. would be even more important for such a study. Again, this study is already very comprehensive with sufficient novelty. Of course, further development is needed for industrial scale operations.

It is quite likely that in an industrial setting, the waste textiles would first be defiberized (like it is done for mechanical recycling) to homogenize the feedstock prior to acid hydrolysis. This would mitigate issues associated with the yarn twist.

3) From my personal view, the details provided in this study are sufficient. Of course, others might disagree with that. But as said, I consider the study already now comprehensive and extensive.

4) Lastly, I don't consider testing extreme compositions (almost pure cotton or pure PET) meaningful in this context. If one can deal with almost pure textile waste fraction, then other recycling strategies would be chosen anyways. The goal of this work is to develop a process that can valorize both constituents in a cotton-polyester blend. Even if pure materials would enter the process stream, the ratio would be adjusted accordingly with other waste textiles. It is also clear that this process would not be a standalone operation, but it would be one possible link in the textile recycling chain, positioned somewhere between textile waste sorting and spinning of new textile fibers.

As said earlier, I believe the quality and novelty of this manuscript justify being published. I would have considered Nature Sustainability as better platform, but also Nature Communication will be an excellent journal to disseminate this very good study and reach the right readership.
